# Effect of Individual Rate of Inbreeding, Recent and Ancestral Inbreeding on Wool Quality, Muscling Conformation and Exterior in German Sheep Breeds

**DOI:** 10.3390/ani13213329

**Published:** 2023-10-26

**Authors:** Cathrin Justinski, Jens Wilkens, Ottmar Distl

**Affiliations:** 1Institute of Animal Breeding and Genetics, University of Veterinary Medicine Hannover (Foundation), 30559 Hannover, Germany; cathrin.justinski@tiho-hannover.de; 2vit—Vereinigte Informationssysteme Tierhaltung w.V., Heinrich-Schröder-Weg 1, 27283 Verden, Germany; jens.wilkens@vit.de

**Keywords:** inbreeding coefficient, ancestral inbreeding, breeding objectives, animal model, genetic parameters, genetic diversity, breeding directions

## Abstract

**Simple Summary:**

The issue of whether increasing inbreeding may be associated with a decrease in the trait values used to select sheep was investigated in all sheep breeds in Germany to obtain a picture of all populations in this country. We analyzed heritabilities and inbreeding depression for the three traits of wool quality, muscling conformation and exterior, for which all sheep intended for breeding are evaluated and upon which breeding decisions in Germany are based. All sheep breeds with sufficient data were included, resulting in 30 different breeds representing all breeding directions in different ecosystems. Heritabilities were across all breeds of moderate size. We estimated the degree of inbreeding depression by animal models and employing linear regressions of the phenotypic trait on inbreeding coefficients calculated from pedigree data. Linear regression slopes were estimated for the individual rate of inbreeding and new and ancestral inbreeding. Inbreeding depression was significant for all three traits when averaged across all 30 sheep breeds and was associated with a reduction in the mean value of the phenotypic trait by 0.33% standard deviation units. Positive effects through ancestral inbreeding seem to have an impact on the exterior of sheep. The results of this study should help to determine the effects of inbreeding on breeding objective traits in sheep breeds.

**Abstract:**

This study provides comprehensive results on the current status of inbreeding depression for traits upon which sheep are selected for the herdbook in Germany. A total of 30 sheep breeds from the OviCap national database met the inclusion criteria for the present analysis regarding the depth and completeness of pedigrees and the number of animals with phenotypic data. We analyzed heritabilities and inbreeding depression for the three breeding objective traits of wool quality, muscling conformation and exterior. Heritabilities were across all breeds of moderate size, with estimates of 0.18 for wool quality and muscling conformation and of 0.14 for exterior. The models employed to estimate linear regression slopes for individual and ancestral inbreeding rates also account for non-genetic effects and the additive genetic effect of the animal. Inbreeding depression was obvious for all three traits when we averaged the estimates across all 30 sheep breeds. Inbreeding depression was significant for wool quality for only a few breeds, whereas for muscling conformation, 14/30 breeds achieved significant estimates. A 1% increase in inbreeding decreased the mean of all three traits across all sheep breeds by 0.33% of their standard deviation. Positive effects due to ancestral inbreeding were only significant in very few breeds in the different traits. Across all 30 sheep breeds, there were indications that purging effects (a reduction in negative effects of inbreeding depression due to selection for heterozygotes) may play a role for the exterior. The results of this study should help for reviewing breeding programs, particularly for sheep breeds with critical effective population sizes and increasing rates of inbreeding, with regard to the selection policy and selection intensity applied.

## 1. Introduction

During the long history of domestication of livestock, breeds were developed to meet the needs for the human food supply and to be adapted to the local environmental and climatic conditions. Among livestock species, most breeds can be considered as populations of limited size, because a limited number of reproductive partners contribute unequally to the next generation, and the selected reproductive partners are likely to be related in a quantitative scheme [1]. Inbreeding is an emerging issue for breed conservation as well as for large commercial populations because of its negative effects on traits of interest. Inbreeding depression has been widely documented for livestock, wild species, model organisms and plant populations. It is generally believed that inbreeding has a greater negative effect on fitness traits than on other traits [2]. In addition to Leroy [3], Doekes [2] compiled 154 different studies across all sorts of traits including seven livestock species and concluded that an increase in pedigree inbreeding had similar unfavorable effects on all different types of traits. A summary of the results for 33 sheep breeds from the meta-analysis of Doekes [2] is given in Appendix A. An increase in pedigree inbreeding of 1% resulted in a mean decrease in the phenotypic value by 0.35% of the trait mean. This outcome and the results of our previous study of genetic diversity of 35 different sheep breeds in Germany [4] gave us reason to examine the hypothesis that there is no statistically defensible evidence that any mammalian species is not affected by inbreeding [5], even in the variety of breeds common here. Inbreeding, which increases over generations, cannot be prevented in populations of a given population size [6] and allows for the influence by inbreeding depression of any trait subject to selection. Thus, the assumption that inbreeding is causally responsible only for a reduction in the mean phenotypic value of traits related to reproductive ability or physiological performance is refuted [7]. Since the magnitude of inbreeding depression is thought to vary among breeds, because populations do not share the same demographic and selection history [8], it is useful to compare different breeds and populations. In order to investigate and further explore the assumption of a varying extent of inbreeding depression in different breeds, the present work will follow up on the results of the previous study and provide insights into the effect of inbreeding depression on traits for which sheep are selected in Germany. The present study should extend previous work through the comparison of many breeds selected for different purposes and under a great variety of different ecosystems. In addition to inbreeding depression due to classical inbreeding coefficients, ancestral inbreeding coefficients and purging effects will be analyzed [9,10,11].

Unlike Doekes’ comparative study [2] on inbreeding depression for different traits and livestock species, this work focuses on all sheep breeds in Germany with sufficient pedigree and performance data recorded for herdbook entry. Very few previous studies have reported traits similar to those that we study here. For example, Ercanbrack and Knight [12] demonstrated a decrease in fleece and body weight as a significant effect of inbreeding in the Targhee and Rambouillet breeds. Similar effects were also observed in the Indian sheep breed Muzaffarnagari [13]. For six linear body traits, body condition and body weight, significant effects of inbreeding depression were found in three different Scottish sheep breeds [14,15].

Despite breeding objectives to further improve performance, breed-typical traits and genetic diversity should be maintained at the same time. The results of our last study encouraged us to perform a follow-up study on inbreeding depression in sheep breeds in Germany, particularly as many sheep breeds with different breeding and utilization directions have developed due to the diverse environmental and climatic conditions. For each of the 30 breeds in this follow-up study, results were available on the traits of wool quality, muscling and exterior, so our results again reflect a picture of the situation in German sheep breeding.

There is a need for studies that investigate the issue of inbreeding purging and the effects of inbreeding on the adaptability of livestock species [3]. Since it is assumed that ancestral inbreeding is less harmful than new inbreeding [16], we address this topic in our present study on 30 sheep breeds in Germany. As individuals with inbred ancestry are less likely to be carriers of deleterious alleles compared to individuals with recent inbreeding [3], and ancestral inbreeding has shown increasing trends in sheep breeds in Germany, the question as to whether purging effects may be found in a larger number of different breeds may be of particular interest [4]. Based on a simulation study in Jersey cattle, Gulisija and Crow concluded [17] that inbreeding purge can be efficient only for alleles with a large impact, reducing the genetic load for alleles with a strong impact. Conversely, purging would likely be negligible for alleles with low impact. Depending on the extent of inbreeding and inbreeding depression, the probability of purging depends to a large extent on the genetic basis of the population in question, which requires research into this topic in many species [18]. A real-world example of this theory is shown in a work on inbreeding purge for milk production traits in Holstein cattle, with a greater proportion of the inbreeding depression being due to new inbreeding [8]. Different hypotheses attempting to explain this phenomenon have been proposed, for example, by Kristensen and Sorensen [19]. Once the underlying mechanisms of purging are clarified, this should be understood as a tool of long-term sustainable livestock production.

The main objectives of the present study were to estimate genetic parameters for wool quality, muscle conformation and exterior (visual appearance of specific body parts) and to quantify the extent of inbreeding depression. In addition, we examine whether negative variants may be eliminated through selection, resulting in positive effects associated with ancestral inbreeding coefficients. The present work thus allows for an overall assessment of the influence of the breeding management style practiced in the different sheep breeds from Germany and the resulting consequences with regard to inbreeding depression and purging.

## 2. Materials and Methods

As a follow-up study to the earlier work, the data sets used in this paper were extracted from serv.it OviCap by Vit/Verden, Germany. Vit is the leading information service provider for animal husbandry and breeding. With OviCap, the service company Vit makes data from one source available to livestock farmers and breeders. The sheep breeds studied here were selected according to the depth of the pedigrees, resulting from the number of complete equivalent generations (GE), and the number of phenotypic records available. Only breeds with a GE > 3.5 and more than 2000 phenotypic records per trait were considered, resulting in 30 sheep breeds (Table 1). As amounts of phenotype data and individuals available for each breed differ from our previous study, GE was recalculated for these data and thus differs slightly from the values in the previous report. Like in the previous study [4], the selection of breeds focused on all breeding and utilization directions as well as autochthonous and imported breeds. According to German legislation, autochthonous breeds are breeds that were first established in Germany or are only bred in Germany, or they are breeds with an independent breeding program in Germany. The inbreeding coefficients according to Meuwissen and Luo [20], Ballou [21] and Kalinowski et al. [22] were retrieved for the animals included in the present study from our previous analysis, which was performed using PEDIG [23] and GRAIN, version 2.2 [24] (Appendix A).

According to Ballou, the ancestral inbreeding coefficient (F_a_Bal_) captures the proportion of the genome of an individual previously exposed to ancestral inbreeding [21]. Therefore, inbreeding in each ancestor is taken into account and summed up over all ancestors. A gradual increase in F_a_Bal_ compared to F over generations implies an increase in inbred ancestors in the pedigrees, but not necessarily an increase in the individual inbreeding coefficient.

According to Kalinowski et al. [22], ancestral inbreeding (F_a_Kal_) means that homozygous alleles have already met in former generations, whereas new inbreeding (F_a_New_) is caused by alleles that are identical-by-descent (IBD) for the first time in this pedigree. F_a_Kal_ is zero if classical inbreeding is zero, because common ancestors have to be present on both sides of the pedigree to be able to calculate an ancestral inbreeding coefficient F_a_Kal_.

Ballou’s concept of inbreeding regards all ancestors in the pedigree, unlike its contribution to the individual inbreeding, which results in an increase in F_a_Bal_ with an inbred ancestor in each generation of a pedigree. A faster increase in F_a_Bal_ compared to F across generations indicates an accumulation of inbred ancestors in the pedigrees, but these inbred ancestors do not necessarily contribute to the individual inbreeding coefficient.

Purging reduces the negative effects of inbreeding depression, because many deleterious alleles express their harmful effects only when homozygous. Reducing the frequency of deleterious alleles in ancestral generations by natural or artificial selection after exposing them to inbreeding leads to purging. Purging is quantified by estimating the linear regression slope of ancestral inbreeding coefficients on the phenotypic trait. Positive estimates for ancestral inbreeding coefficients indicate the presence of purging.

The responsibility for the performance tests of German herdbook animals lies with the breeding association and is carried out as a field test, either compulsory or voluntary depending on the breed, by trained personnel (commission of three experienced persons, including the head of breeding board) of the respective association. Overall scores for each of the three traits of wool quality, muscling conformation and exterior are recorded for all breeds, for both male and female animals intended for breeding, within the framework of the herdbook entry according to the regulations in the herdbook. The scores awarded by the commission, on a scale from 1 to 9, are valid for the whole life and serve as the basis for the classification of animals by their breeding value and the decision for herdbook entry. A minimum score of 4 must be achieved for each trait for herdbook admission. A score of 9 for wool quality, muscling and exterior is the optimum for an animal of each breed and corresponds best to the breeding objectives. The final scores of all three breeding objective traits and all animals presented at the field test are recorded in OviCap. Therefore, the whole variation in these final scores from 1 to 9 is captured for each breed. The final scores for wool quality, muscling conformation and partly for exterior (such as breed-typical pigmentation, body size and head shape) are breed-specific. They are thus centered on the breed average and are therefore not comparable between breeds. The evaluation takes place in selected locations for a group of animals from different farms. For the assessment of wool quality, a score from 1 to 9 is given for the assessed criteria including fineness, color, length, density, balance and growth at the shoulder, chest and haunches. It is then summarized with equal weights in a final score. In the case of typical wool defects such as stalky, matted or yellow-sweaty wool, twist, color defects or stitch hairs, the final score is downgraded to 1–4 depending on the grade and extension of the wool abnormality. Instead of the wool quality, the shedding behavior during molting and the coat quality (color, shine, structure, balance) of hair sheep are evaluated; just as the wool quality, a score from 1 to 9 is given and combined for a final score (Appendix A). The evaluation should be performed after the first wintering of the animals.

For muscling conformation, the flesh-bearing parts of the chest, shoulder, back and haunch are evaluated and then combined into a final muscling score, with the back and haunch weighted higher. The most important single criteria for exterior include head shape, dentition, horn status, head pigmentation and breed-typical appearance; pigmentation of the wool, legs and head; height at the withers, body length, rump depth and width of the breast; conformation of the neck and shoulder; backline and conformation of the back; length, width and inclination of the pelvis; angle and position of the carpal and hock joints; angle at the pasterns, spread of the claws and movement. The scores of the single criteria are summarized with equal weights in a final score if no defects in the body conformation are observed. In the case of severe defects in body conformation such as underbite, kyphosis, lordosis, broken pasterns or steep ankle posture or signs of disorders of sexual development like testicles that are too small or split, the final score is downgraded to less than 4 in order to exclude animals from breeding.

In order to further investigate breeds with similar breeding objectives, we assigned the 30 sheep breeds to seven different breeding directions [4] with merinos, meat, country, mountain/stone, heath, milk and exotic breeds (Appendix A). Merino sheep have the finest wool and are fast-growing, with medium size. Meat sheep are medium- to large-framed, fast-growing and well muscled at the chest, shoulder, rump, back and haunch; they have also good wool quality. Country sheep are well adapted to the ecosystem in which they evolved. Fineness and type of wool as well as muscling conformation are adapted to the local conditions. These sheep are undemanding, capable of marching and hardy, with a stable foundation and good claw health. Mountain and stone sheep are adapted to the harsh mountain conditions in South Germany and have similar characteristics as country sheep. Another specific group of country sheep are heath sheep. The heath breeds are small- to medium-framed animals that are indispensable for the landscape management on the heath, and they are particularly characterized by their frugality and robustness. They serve as versatile meat and wool suppliers and improve the fertility of the heath soils through their manure. The meat yield is very low, as they are frugal, but the meat has a wildlife flavor. The dairy sheep considered in this study are large-framed East Friesian dairy sheep with moderate muscling and coarse wool. The exotic breeds are phenotypically rather different breeds such as hair sheep and small-sized sheep. Endangered breeds are mainly found under country, mountain/stone and heath sheep breeds (Appendix A). For the 30 sheep breeds, we estimated additive genetic (co-)variances and residual (co-)variances as well as heritabilities, genetic and residual correlations. The final data set for the 30 breeds contained 169,414 animals with scores for all three traits: wool quality, muscling conformation and exterior. The median number of animals per breed was 3018.5, with first and third quartiles of 1526 and 8394 individuals, respectively. The following linear multivariate animal model, parameterized according to the models employed in the routine evaluation of breeding values through Vit/Verden, was employed:Y_ijklm_ = µ + LOC-YEAR_i_ + SEX-MULTIPLES_j_ + AGE_k_ + SEASON_l_ + animal_m_ + e_ijklm_, (1)
where Y_ijklm_ = overall score for wool quality, muscling conformation and exterior; LOC-YEAR_i_ is the ith location by year effect for each breed, with different numbers of levels for each breed; SEX-MULTIPLES_j_ is the jth class of the sex of the animal by number of animals born per lambing for j = 1–2 (sex) by 1–4 (up to four multiples, depending on sheep breed); AGE_k_ = age at scoring in days by six classes for k = 1 (60–129), 2 (130–159), 3 (160–189), 4 (190–365), 5 (360–729) and 6 (730–1095); SEASON_l_ is the lth class of the lambing season for l =1 (December–May) and 2 (June–November) for seasonal breeds otherwise omitted; animal_m_ = random animal effect; and e_ijklm_ = random error term.

The effects of the different inbreeding coefficients were calculated by extending model 1 with linear regressions (b_1_ to b_5_) on the respective inbreeding coefficients. We employed three differently parameterized models for analyzing the effects of inbreeding coefficients. Model 2 regarded the individual rate of inbreeding (ΔF):Y_ijklmn_ = µ + LOC-YEAR_i_ + SEX-MULTIPLES_j_ + AGE_k_ + SEASON_l_ + b_1_ΔF_m_ +animal_n_ + e_ijklmn_


In model 3, the effects of the ancestral (F_a_Kal_) and new inbreeding coefficient (F_New_) according to Kalinowski et al. (2000) [22] were simultaneously analyzed:Y_ijklmno_ = LOC-YEAR_i_ + SEX-MULTIPLES_j_ + AGE_k_ + SEASON_l_ + b_2_F_a_Newm_ + b_3_F_a_Kaln_ + animal_o_ + e_ijklmno_


In model 4, the classical inbreeding coefficient (F) and the interaction of F_a_Bal_ with F, as suggested by Ballou (1997) [21], were simultaneously considered as the probability that an individual is identical by descent (IBD), which is itself is not taken into account by F_a_Bal_:Y_ijklmno_ = LOC-YEAR_i_ + SEX-MULTIPLES_j_ + AGE_k_ + SEASON_l_ + b_4_F_m_ + b_5_F × F_a_Baln_ + animal_o_ + e_ijklmno_


Variance and covariance components were estimated with model 1 and using VCE 6.0.2 [25]. For models 2–4, we used estimated additive genetic and residual (co-)variances to estimate effects for linear regressions using PEST, version 4.2.6 [26]. Further statistical analyses were performed in SAS, version 9.4 (Statistical Analysis System, Cary, NC, USA, 2023).

## 3. Results

We analyzed overall scores for the wool quality, muscling conformation and exterior of 30 sheep breeds (Table 1). For each breed, the breeding direction, size of the pedigree, GE, means and standard deviations and the heritabilities for the traits are given.

### 3.1. Estimates for Heritabilities, Residual and Genetic Correlations and (Co-)Variances

The mean (median) heritability for wool quality was 0.18 on average across all breeds (Appendix A, Appendix A). The values of the first and third quartile were 0.14 and 0.20 and included 7/30 and 22/30 breeds, respectively. The SWS, WBS and DOS breeds showed the highest estimates, with 0.34 ± 0.08, 0.37 ± 0.03 and 0.48 ± 0.04, respectively, whereas Ouessant and Bentheim showed the lowest estimates, with 0.03 ± 0.04 and 0.03 ± 0.02.

The mean heritability of muscling conformation across all breeds was also 0.18, and we found two outliers: Ouessant, with a maximum value of 0.34 ± 0.07, and Carinthian, with a minimum value of 0.02 ± 0.09 (Appendix A, Appendix A). The 25% quartile was 0.14 and contained 9/30 breeds, whereas the 75% quartile, with a value of 0.22, contained 23/30 breeds.

The mean heritability for the exterior trait was 0.14, and all but two values were in the range between the upper and the lower quartile. The breed with the lowest estimate was Swifter, with an estimate of 0.02 ± 0.03. The highest value was achieved by Berrichon du Cher, with 0.27 ± 0.08 (Appendix A, Appendix A). The first quartile was 0.11 and included 7/30 breeds, and the third quartile (h^2^ = 0.18) contained 22/30 breeds.

### 3.2. Inbreeding Depression for Wool Quality

#### 3.2.1. Individual Rate of Inbreeding

A significant negative regression was found for GGH (−6.19 ± 1.74), SKU (−5.17 ± 1.84), MLW (−4.16 ± 1.73), WKF (−3.24 ± 1.52) and CHA (−2.82 ± 1.21), and a significant positive regression was found for NOL (8.58 ± 3.38) (Appendix A). The RHO breed was close to the significance threshold, with a *p*-value of 0.0505 for the estimate of −1.59 ± 0.97.

#### 3.2.2. Ancestral and New Inbreeding according to Kalinowski

The regressions of the ancestral inbreeding coefficient F_a_Kal_ on wool quality were significantly negative for BDC (−19.14 ± 11.05) and LES (−2.59 ± 1.35). MLS showed a positive value of 1.31 ± 0.82, and this result was close to the significance threshold. For the new inbreeding coefficient F_a_New_, significant negative results could be detected for CHA (−2.24 ± 0.82), SKU (−2.22 ± 0.70), MLW (−1.29 ± 0.50) and WAD (−1.09 ± 0.41), and an almost significant negative value was found for AST (−0.87 ± 0.54). Nolana was the only breed with a significant positive regression coefficient (3.00 ± 1.68) (Appendix A).

#### 3.2.3. Interaction of F × F_a_Bal_ and F

The average of F across all breeds gave a significant negative result (−0.2454 ± 0.1148). The interaction of F × F_a_Bal_ was significantly negative for BDC, MFS and LES (−39.72 ± 24.06, −9.50 ± 5.06 and −7.12 ± 2.84). The WAD, SKF and OMS breeds showed significant positive results (5.90 ± 3.26, 6.18 ± 3.40, 7.10 ± 4.47) (Appendix A).

### 3.3. Inbreeding Depression for Muscling Conformation

#### 3.3.1. Individual Rate of Inbreeding

For the individual rate of inbreeding, 14/30 breeds (MLW, SKU, MFS, GGH, WBS, WKF, WHH, COF, BRI, SKF, RHO, MLS, TEX and IDF) showed significant negative regression coefficients (Appendix A).

#### 3.3.2. Ancestral and New Inbreeding according to Kalinowski

For F_a_Kal_, only TEX (−2.38 ± 0.93) reached a significant negative value, and only WHH (2.34 ± 1.18) a significant positive value. For F_a_New_, all but two breeds (WGH, NOL) had a negative regression coefficient, and it was significantly negative in 7/30 breeds (WHH, MLW, IDF, LES, RHO, MFS and SKF) (Appendix A).

#### 3.3.3. Interaction of F × F_a_Bal_ and F

All breeds except NOL showed negative regression coefficients for F, with 14/29 breeds (MLW, SKU, MFS, IDF, AST, GGH, COF, TEX, WKF, RHO, WHH, SKF, SUF and MLS) being significantly negative. The across-breed average was significantly negative. The interaction of F × F_a_Bal_ was significantly negative for TEX (−6.08 ± 3.69), and WHH was the only breed with a significant positive regression (5.20 ± 2.99) (Appendix A).

### 3.4. Inbreeding Depression for Exterior

#### 3.4.1. Individual Rate of Inbreeding

For 11/30 breeds (SWS, LES, OMS, MLW, COF, BRI, WAD, SKF, IDF, MLS and SUF), significant negative estimates were obtained for ΔF_i_ (−1.72 to −3.03) (Appendix A).

#### 3.4.2. Ancestral and New Inbreeding according to Kalinowski

F_a_Kal_ showed a significant negative regression coefficient in only 2/30 breeds (GGH, TEX). MLW was the only breed with a significant positive estimate. For F_a_New_, 12/30 breeds (SWS, DOS, WKF, MLW, OMS, WAD, IDF, MFS, SUF, RHO, SKF and COF) had significantly negative estimates (Appendix A).

#### 3.4.3. Interaction of F × F_a_Bal_ and F

Significant negative regression coefficients for F were estimated in 11/30 breeds (SWS, IDF, LES, WAD, MLW, AST, SUF, COF, TEX, SKF and MLS) (Appendix A). The interaction of F × F_a_Bal_ was significantly negative in GGH (−11.97 ± 5.38) and TEX (−7.62 ± 4.55). A significant positive regression coefficient was obtained for OMS (6.34 ± 3.38).

### 3.5. Inbreeding Depression across Breeds

Thus, a further objective of the present study was to quantify the degree of inbreeding depression across all 30 sheep breeds and, in addition, to examine whether negative variants may be eliminated through selection, resulting in positive effects associated with ancestral inbreeding coefficients (Table 2).

Overall means for all 30 breeds under study showed significant negative slopes for the regression of wool quality, muscle conformation and exterior on ΔF_i_, F and F_a_New_, whereas significant positive effects on ancestral inbreeding F_a_Kal_, F × F_a_Bal_ were not found. In order to be able to compare the present results with previous reports, estimates of regression coefficients were scaled to the respective phenotypic (b_σp_) and additive genetic standard deviations (b_σa_) of the traits using variance components from model 1. For wool quality, muscle conformation and exterior, median per 1% increase in F was −0.33% (CI25–75%: −0.73 to −0.06, skewness = −0.41, kurtosis = 5.11), −0.33% (CI25–75%: −0.71 to −0.06, skewness = 0.29, kurtosis = 5.90) and −0.33% (CI25–75%: −0.72 to −0.06, skewness = 1.15, kurtosis = 6.90), respectively.

Differences between the breeding directions for the decrease per 1% increase in inbreeding were not significantly different for any of the three traits. However, for heath sheep, the median decreases due to inbreeding depression expressed with b_σp_ for F were largest among the breeding directions with values of −1.21% (CI25–75%: −1.51 to −0.48), −1.19% (CI25–75%: −1.51 to −0.47) and −1.19% (CI25–75%: −1.54 to −0.48) for wool quality, muscle conformation and exterior, respectively. Using the median decrease scaled to the standard deviation (b_σp_) of the wool quality, muscle conformation and exterior score per 1% increase in ΔF_i_, the resulting estimates across all breeds were −1.29% (CI25–75%: −2.33 to −0.23, skewness = 0.01, kurtosis = 2.59), −1.29% (CI25–75%: −2.25 to −0.25, skewness = 0.52, kurtosis = 3.84) and −1.25% (CI25–75%: −2.32 to −0.25, skewness = 0.80, kurtosis = 5.03), respectively.

### 3.6. Inbreeding Depression for Wool Quality by Breeding Directions

We compared the estimates for the regression coefficients between breeding directions and found significant differences for ΔF_i_, F_a_New_ and F. These significant differences are due to the breeding directions HEA and EXO.

#### 3.6.1. Individual Rate of Inbreeding

Compared in the different breeding groups, three of six (CON, MON and HEA) showed significant negative regression coefficients (Appendix A).

#### 3.6.2. Ancestral and New Inbreeding according to Kalinowski

For the inbreeding coefficients according to Kalinowski, only the heath sheep (HEA) showed a significant result for F_a_New_ (Appendix A).

#### 3.6.3. Interaction of F × F_a_Bal_ and F

Significant negative results for F were estimated for CON and MON (Appendix A).

### 3.7. Inbreeding Depression for Muscling Conformation by Breeding Directions

Significant differences between breeding directions were not found.

#### 3.7.1. Individual Rate of Inbreeding

For four of six breed groups (MER, MEA, CON and HEA), significant negative regression coefficients were obtained (Appendix A).

#### 3.7.2. Ancestral and New Inbreeding according to Kalinowski

For F_a_Kal_, only meat sheep (MEA) achieved a significant negative estimate. Meanwhile, the breeding directions MEA, CON and MON showed significant negative values for F_a_New_ (Appendix A).

#### 3.7.3. Interaction of F × F_a_Bal_ and F

The inbreeding coefficient F showed significant negative regression coefficients for the breeding directions MEA, CON, MON and HEA (Appendix A).

### 3.8. Inbreeding Depression for Exterior by Breeding Directions

Significant differences between breeding directions for F_a_New_ and F were due to MEA.

#### 3.8.1. Individual Rate of Inbreeding

All breeding directions but EXO showed significant negative estimates (Appendix A).

#### 3.8.2. Ancestral and New Inbreeding according to Kalinowski

For F_a_New_, significant negative estimates were determined for the breeding directions MEA and CON (Appendix A).

#### 3.8.3. Interaction of F × F_a_Bal_ and F

The inbreeding coefficient F showed significant negative regression coefficients for the breeding directions MEA, CON and HEA (Appendix A).

## 4. Discussion

The results of our last study on the population genetics and genetic diversity of 35 different sheep breeds bred in Germany revealed evidence of genetic bottlenecks. For half of the breeds, increasing trends in the inbreeding coefficient were found, and in nearly all breeds, significant positive trends in ancestral inbreeding were shown [4]. Therefore, we deemed it necessary to analyze to what extent the phenotypic means of the different breeds are influenced by increasing inbreeding coefficients. Furthermore, there has not been any study analyzing purging effects in sheep. Since inbreeding may be assumed to be due to deleterious alleles when in a homozygous state, it is important to measure the effects of inbreeding on fitness and productive traits as well [27,28]. A consequence of inbreeding may be inbreeding depression, which reduces performance in growth, production, health, fertility and survival [29]. Recent studies suggest that its impact on individual fitness is even greater than previously thought [30]. Inbreeding depression in livestock is estimated for individual performance data using linear regression on the individual pedigree inbreeding coefficient [31]. In the present study, the individual rate of inbreeding ΔFi, developed by Gutierrez et al. (2009) [32], was employed to account for differences in GE between breeds.

The across-breed median for inbreeding depression of −0.33% per 1% increase in F, scaled to the phenotypic standard deviation, was deemed less harmful than the overall median of −0.52% (CI25–75%: −1.43 to −0.04, skewness = 4.26, kurtosis = 47.36) for b_σp_, calculated for F from the meta-analysis of Doekes for sheep (Appendix A), as well as the median across species and traits of −0.59% [2]. Using only the trait categories weight/growth and reproduction for the sheep data of the meta-analysis [2], we found estimates for the median at −0.73% (CI25–75%: −1.38 to −0.27, skewness = −2.66, kurtosis = 6.81) and −0.38% (CI25–75%: −1.99 to 0.08, skewness = 3.46, kurtosis = 20.43), respectively. For conformation, no data were available. The corresponding across-breed estimates of the median for inbreeding depression per 1% increase in ΔF_i_ were 3.8- to 3.9-fold greater than the median values per 1% increase in F. Therefore, we propose ΔF_i_ as a much more sensitive parameter to identify inbreeding effects compared to F.

Our data seemed only very slightly or not skewed and not so widely distributed in comparison to the meta-analysis of Doekes [2]. A reasonable explanation for these differences in skewness and kurtosis may be that in our study, all 30 breeds were from the same country, traits were on the same scale, the number of traits was low, and phenotypic variances of the traits studied here were similar. We expected similar variances for the final scores, because these traits are centered on breed means and based on the same scale. In addition, selection intensity is equal for the three breeding objective traits for all sheep breeds.

We employed linear regressions in agreement with other authors [33,34], because these are well suited for low to medium levels of inbreeding (below 10–20%). In wild species, selection occurs mainly for fitness traits, whereas in livestock, selection occurs not only for fitness traits, but also for traits related to production [2]. These traits are influenced by inbreeding even more because they are intensively selected. For traits with higher heritabilities, inbreeding depression is expected to be estimated with a higher accuracy. In our study, this influence seems to be negligible due to the very similar size of the heritability estimates for all three traits.

Since the method of phenotype assessment for the selection of breeding animals presented in this study is not comparable with previously published reports, it is difficult to compare the estimated values with those of other breeds. Just as the number of studies for production traits far exceeds that of type traits, this is analogous to the situation of estimating heritabilities, for which far more work was performed than for traits such as wool quality and conformation. For example, Safari has performed studies on genetic correlations and heritabilities of wool parameters (such as fiber diameter, staple length, fleece weight), growth (such as post weaning weight, conformation) and reproductive traits in merino sheep [35,36]. For merino sheep, a number of variance and heritability estimates were also reported for various wool quantity and quality parameters [37]. Heritabilities were also estimated for the Rambouillet breed on broadly comparable parameters [38]. The heritabilities for conformation, weaning and post-weaning from these previous studies [35,36,38] ranged from 0.05 to 0.23 and agreed well with the estimates for muscling conformation in the present study, even if the traits were different. Fiber diameter showed heritabilities of 0.57–0.59 [36], 0.09–0.40 [37] and 0.05–0.22 [38], so the estimates from the present study are still in the same range.

Inbreeding depression is particularly important for traits under selection in breeding [39]. The question is whether inbreeding depression has the same effect on all traits used for selection in breeding programs, or whether there are trait-dependent differences. In a meta-analysis of non-domestic animal populations, DeRose and Roff [39] also reported more inbreeding depression for life history traits (fecundity, survival and development) than for morphological traits (adult body size). In our study, we analyzed morphological traits, and, thus, we might expect more severe inbreeding depression in fertility and survival traits for these 30 sheep breeds following the results of DeRose and Roff [39]. Contrasting results revealed the analysis of the sheep meta-data and estimates across traits and species from Doekes [2], with more inbreeding depression for weight/growth traits than for reproduction/survival traits.

The present work is the first to investigate inbreeding effects in traits upon which the decision to breed directly depends on. Although a number of papers have been published on the subject of inbreeding depression and production traits in sheep breeding, none of the authors have yet investigated all traits used for breeding decisions. For example, Ercanbrack and Knight observed a decrease in fleece weight and body weight as a significant effect of inbreeding in the Targhee and Rambouillet breeds [12]. Similar effects were also observed in the Indian sheep breed Muzaffarnagari [13]. Due to the decreasing importance of wool quality in a large number of German sheep breeds, an increased focus and selection intensity is given to muscling conformation and exterior. This may be reflected in the number of breeds for which we found significant inbreeding depression. There were only a few breeds (4/30) with significantly negative regressions for the individual rate of inbreeding on wool quality, whereas for exterior and muscling conformation, 11/30 and 14/30 reached significantly negative regression coefficients. For traits that are under strong directional selection, such as primary fitness traits like survival, the average dominance effect is expected to be favorable, as fixation is faster for loci with an unfavorable dominance effect [40]. For traits less related to fitness and for traits subject to stabilizing selection, directional dominance is expected to be less pronounced due to lower directional selection pressure [2]. Livestock populations are usually subject to directed selection for a combination of production, conformation, growth, reproduction, survival, behavior and health traits, in addition to natural selection for primary fitness traits. Therefore, all of these trait groups may have similar levels of directional dominance and, consequently, similar levels of inbreeding depression. This is also consistent with the relative dominance variance (i.e., the proportion of dominance variance in phenotypic variance), which appears to be similar across trait groups in livestock [2]. Even though earlier studies are only comparable to this work to a very limited extent, clear indications of the presence of inbreeding depression with regard to various body traits were previously found in Wiener’s work, which examined various body measurements in the three Scottish breeds of Scottish Blackface, Cheviot (South Country) and Welsh Mountain [15]. With increasing inbreeding, Wiener reported a significant reduction in the size of body measurements, with most of the linear and many of the non-linear effects of inbreeding of the individual being significant. Breed-dependent differences in body measurements were also observed by Wiener. As expected for the breeds used in the experiment, the variation attributable to breed was significant (*p*-Value < 0.001) for each of the linear body measurements at all ages studied, and the variation between the three crossbred types was also significant. In agreement with this work [15], a significant inbreeding depression was obvious for F in all three traits.

Recent inbreeding across all sheep breeds was more damaging than ancestral inbreeding in our data. However, a few breeds (MLW, WHH) had significant positive estimates for F_a___Kal_. Thus, purging effects cannot be generally excluded in sheep breeds. Considering the definition of Ballou (1997) [21], a significant positive regression coefficient of the interaction of F with F_a_Bal_ would be evidence for the occurrence of purging. Significant beneficial effects were found in more breeds compared to F_a___Kal_, with 3/30, 1/30 and 2/30 sheep breeds for wool quality, muscle conformation and exterior, respectively. In managed populations, minimizing inbreeding or kinships are the main approaches for minimizing inbreeding depression, based primarily on genealogical information, but possibly also based on detailed genomic data when available [41]. Eliminating inbreeding depression, i.e., reducing the frequency of harmful variants that lead to inbreeding depression in homozygosity, can be achieved either naturally or through planning [18,42]. Then, when fitness-reducing alleles are eliminated, fitness rises again to a higher level, slightly below the original level, as the deleterious alleles are partially fixed [30]. According to Doekes, in order to get to the bottom of the causes of inbreeding depression in different breeds and populations, and thus to ensure sustainable, sensible selection that preserves genetic resources, it seems sensible to use genomic data [43,44,45,46] for these analyses in addition to pedigree-based investigations [2].

## 5. Conclusions

Inbreeding depression across all 30 sheep breeds was obvious for the individual rate of inbreeding, new inbreeding and classical inbreeding coefficients. Effects of inbreeding depression, expressed as a 1% increase in inbreeding per the trait’s standard deviation, were of very similar size for the traits of wool quality, muscle conformation and exterior. Favorable effects through alleles that were identical-by-descent in ancestral generations seem to play a role in the exterior in sheep breeding, and significant effects have been shown in a few breeds. Efforts to maintain genetic diversity in sheep can be made more efficient by taking inbreeding depression into account, particularly in sheep breeds with increasing trends in individual rates of inbreeding. The results of the present study should be useful in evaluating breeding programs and selection intensity, particularly for sheep breeds classified as threatened.

## Figures and Tables

**Table 1 animals-13-03329-t001:** Analyzed sheep breeds, grouped into breeding directions (BD), with the number of animals in the pedigree (N_ped_), the number of complete equivalent generations (GE), the number of studied animals for each trait (N), the means and their standard deviation for exterior, muscling conformation and wool quality, as well as the heritabilities (h^2^) for each trait with the standard deviation.

Code	Breed	BD	N_ped_	GE	N	Exterior		Muscling Conformation		Wool Quality	
						$\bar{X}$ ± SD	h^2^	$\bar{X}$ ± SD	h^2^	$\bar{X}$ ± SD	h^2^
**AST**	Alpine Steinschaf	MON	10,420	4.77	2066	7.26 ± 0.66	0.15 ± 0.04	7.36 ± 0.63	0.22 ± 0.06	7.39 ± 0.70	0.16 ± 0.05
**BBS**	Brown Mountain	MON	22,961	5.97	1962	7.23 ± 0.71	0.20 ± 0.04	7.37 ± 0.61	0.16 ± 0.05	7.22 ± 0.70	0.22 ± 0.03
**BDC**	Berrichon du Cher	EXO	4680	3.67	900	7.49 ± 0.72	0.27 ± 0.08	7.86 ± 0.59	0.17 ± 0.08	7.51 ± 0.66	0.09 ± 0.08
**BLS**	Bentheim	CON	46,173	8.04	5622	7.16 ± 0.69	0.14 ± 0.02	7.47 ± 0.59	0.14 ± 0.03	7.27 ± 0.65	0.03 ± 0.02
**BRI**	Carinthian	CON	10,669	4.17	1159	7.07 ± 0.73	0.04 ± 0.04	7.28 ± 0.61	0.02 ± 0.09	7.11 ± 0.75	0.33 ± 0.08
**CHA**	Charollais	MEA	11,237	3.21	1266	7.40 ± 0.82	0.12 ± 0.11	7.85 ± 0.70	0.09 ± 0.08	7.51 ± 0.70	0.03 ± 0.06
**COF**	Coburg	CON	70,156	7.62	8225	7.42 ± 0.68	0.14 ± 0.02	7.53 ± 0.62	0.17 ± 0.02	7.37 ± 0.73	0.17 ± 0.02
**DOS**	Dorper	MEA	36,057	5.82	810	7.43 ± 0.73	0.19 ± 0.03	7.66 ± 0.71	0.14 ± 0.03	7.57 ± 1.08	0.48 ± 0.04
**GGH**	German Grey Heath	HEA	69,369	7.52	8394	7.16 ± 0.68	0.15 ± 0.05	7.42 ± 0.65	0.18 ± 0.09	7.16 ± 0.70	0.14 ± 0.02
**IDF**	Ile-de-France	MEA	14,021	3.73	2390	7.52 ± 0.71	0.13 ± 0.04	7.74 ± 0.65	0.07 ± 0.03	7.44 ± 0.74	0.17 ± 0.04
**KST**	Krainer Steinschaf	MON	9671	5.51	2098	7.27 ± 0.71	0.15 ± 0.06	7.41 ± 0.62	0.14 ± 0.05	7.49 ± 0.63	0.18 ± 0.04
**LES**	Leine	CON	42,949	7.61	6488	7.31 ± 0.70	0.14 ± 0.02	7.41 ± 0.68	0.20 ± 0.02	7.44 ± 0.71	0.16 ± 0.02
**MFS**	German Mutton Merino	MER	132,413	6.68	10,737	7.21 ± 0.91	0.09 ± 0.02	7.45 ± 0.76	0.16 ± 0.02	7.26 ± 0.90	0.12 ± 0.02
**MLS**	German Merino	MER	204,494	8.42	28,811	7.47 ± 0.73	0.14 ± 0.02	7.69 ± 0.61	0.20 ± 0.02	7.49 ± 0.77	0.16 ± 0.02
**MLW**	Merino Longwool	MER	61,216	6.59	9489	7.22 ± 0.77	0.10 ± 0.02	7.25 ± 0.81	0.18 ± 0.03	7.44 ± 0.78	0.16 ± 0.03
**NOL**	Nolana	EXO	11,920	3.92	748	7.21 ± 1.08	0.17 ± 0.08	7.61 ± 0.79	0.23 ± 0.07	7.40 ± 1.90	0.41 ± 0.10
**OMS**	East Friesian	MIL	71,159	8.20	6597	7.29 ± 0.68	0.14 ± 0.12	7.36 ± 0.64	0.21 ± 0.21	7.29 ± 0.64	0.21 ± 0.11
**OUS**	Ouessant	EXO	10,051	6.47	1452	7.39 ± 0.69	0.14 ± 0.07	7.33 ± 0.51	0.34 ± 0.07	7.51 ± 0.61	0.03 ± 0.04
**RHO**	Rhön	CON	78,095	6.22	8931	7.35 ± 0.79	0.10 ± 0.02	7.46 ± 0.71	0.23 ± 0.02	7.51 ± 0.76	0.14 ± 0.02
**RPL**	Pomeranian Coarsewool	CON	56,965	7.47	2091	7.28 ± 0.70	0.07 ± 0.02	7.33 ± 0.59	0.06 ± 0.02	7.31 ± 0.77	0.15 ± 0.04
**SKF**	German Blackhead Mutton	MEA	128,839	7.80	18,921	7.45 ± 0.71	0.12 ± 0.01	7.67 ± 0.62	0.15 ± 0.01	7.44 ± 0.68	0.16 ± 0.01
**SKU**	Skudde	HEA	32,747	6.50	1526	7.10 ± 0.76	0.25 ± 0.30	7.28 ± 0.58	0.23 ± 0.16	7.29 ± 0.73	0.15 ± 0.36
**SUF**	Suffolk	MEA	68,136	5.25	11,744	7.38 ± 0.80	0.19 ± 0.01	7.68 ± 0.65	0.23 ± 0.01	7.37 ± 0.72	0.15 ± 0.01
**SWS**	Swifter	MEA	4608	3.71	395	7.24 ± 0.74	0.02 ± 0.03	7.63 ± 0.57	0.12 ± 0.07	7.33 ± 0.59	0.34 ± 0.08
**TEX**	Texel	MEA	58,223	5.71	9288	7.48 ± 0.75	0.14 ± 0.01	7.81 ± 0.61	0.20 ± 0.01	7.46 ± 0.64	0.20 ± 0.01
**WAD**	Wald	CON	17,172	5.65	1883	7.21 ± 0.70	0.21 ± 0.06	7.36 ± 0.66	0.13 ± 0.06	7.29 ± 0.73	0.05 ± 0.04
**WBS**	White Mountain	MON	30,188	8.16	3118	7.23 ± 0.70	0.19 ± 0.03	7.56 ± 0.62	0.30 ± 0.04	7.38 ± 0.76	0.37 ± 0.03
**WGH**	German White Heath	HEA	18,158	7.37	2919	7.11 ± 0.68	0.11 ± 0.04	7.42 ± 0.59	0.27 ± 0.05	7.35 ± 0.61	0.18 ± 0.04
**WHH**	White Polled Heath	HEA	41,306	8.69	5585	7.23 ± 0.64	0.05 ± 0.02	7.44 ± 0.61	0.17 ± 0.03	7.42 ± 0.59	0.11 ± 0.03
**WKF**	German Whitehead Mutton	MEA	38,390	7.52	3799	7.27 ± 0.73	0.18 ± 0.07	7.57 ± 0.61	0.15 ± 0.06	7.45 ± 0.62	0.18 ± 0.04

Abbreviations for breeding directions: country: CON, exotic: EXO, heath: HEA, meat: MEA, merino: MER, milk: MIL, mountain/stone: MON.

**Table 2 animals-13-03329-t002:** Animal model linear regression coefficients of the inbreeding depression derived from the individual rate of inbreeding (ΔF_i_), the ancestral (F_a___Kal_) and new (F_a___New_) inbreeding coefficient according to Kalinowski, inbreeding coefficient (F) and interaction between F and the ancestral inbreeding coefficient according to Ballou (F × F_a_Bal_) on the final score of exterior, muscling conformation and wool quality, with their corresponding standard deviations (SD), standard errors (SE), the 95% and 5% confidence intervals (95% CI, 5% CI) and their *p*-Values across all 30 breeds studied.

Model			Exterior	Muscling Conformation	Wool Quality
**2**	ΔF_i_	Mean	−2.0354	−1.9616	−0.9675
		SD	2.4400	1.5191	2.4733
		SE	0.4455	0.2773	0.4515
		95% CI	0.9515	−0.18460	1.3985
		5% CI	−6.8340	−4.1333	−5.1689
		*p*-Value	<0.0001	<0.0001	0.0407
**3**	F_a___Kal_	Mean	0.3279	−1.2755	0.0630
		SD	1.7987	2.1780	4.1744
		SE	0.3284	0.3976	0.7621
		95% CI	4.2072	1.5615	5.1965
		5% CI	−2.8503	−4.3568	−2.6005
		*p*-Value	0.3264	0.0033	0.9346
**3**	F_a___New_	Mean	−0.6834	−0.4444	−0.3705
		SD	0.8504	0.4846	0.9229
		SE	0.1553	0.0884	0.1685
		95% CI	0.3830	0.3893	0.5022
		5% CI	−2.1012	−1.2420	−2.2237
		*p*-Value	0.0001	<0.0001	0.0360
**4**	F	Mean	−0.5325	−0.5185	−0.2454
		SD	0.7634	0.3386	0.6290
		SE	0.1394	0.0618	0.1148
		95% CI	0.0982	−0.0794	0.1937
		5% CI	−1.2071	−1.0735	−1.0283
		*p*-Value	0.0007	<0.0001	0.0412
**4**	F × F_a_Bal_	Mean	2.7898	−3.7863	1.5953
		SD	8.5078	9.5145	12.2412
		SE	1.5533	1.7371	2.2349
		95% CI	14.8244	5.2029	18.8205
		5% CI	−7.6672	−34.2169	−9.5945
		*p*-Value	0.0829	0.0375	0.4811

## Data Availability

Restrictions apply to the availability of these data. Data were obtained from vit/Verden and are available on a reasonable request from the authors with the permission of the German sheep breeding associations.

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
