# Peer review of "Effect of Individual Rate of Inbreeding, Recent and Ancestral Inbreeding on Wool Quality, Muscling Conformation and Exterior in German Sheep Breeds"

_animals, 2023, doi:10.3390/ani13213329_

Round 1

Reviewer 1 Report

Comments and Suggestions for Authors

 Effect of individual rate of inbreeding, recent and ancestral in-2 breeding on wool quality, muscle conformation and exterior in 3 German sheep breeds

Keywords: inbreeding depression; sheep breeds; production traits; wool quality; muscling confor-56 mation; exterior; purging

There are words in the title as keywords. You can't repeat them. You must choose just one plaice.

The abstract is very long and repetitive. The authors could be more objective and reduce this to a maximum of 250 words.

The paper does not have a clear and defined objective.

Why were the objectives included in the Discussion? And not at the end of the introduction?

Line 353-355: Thus, the main objectives of the present study were to quantify the degree of inbreeding depression and in addition, whether negative variants may be eliminated through selection resulting in positive effects associated with ancestral inbreeding coefficients.

Comments on the Quality of English Language

All work should be written in the third person. In some places, the authors say we (our)

Author Response

Reviewer 1

We thank the reviewer for the valuable and useful comments in order to improve our manuscript.

We amended our manuscript accordingly.

Yes

Can be improved

Must be improved

Not applicable

Does the introduction provide sufficient background and include all relevant references?

(x)

( )

( )

( )

Are all the cited references relevant to the research?

(x)

( )

( )

( )

Is the research design appropriate?

(x)

( )

( )

( )

Are the methods adequately described?

(x)

( )

( )

( )

Are the results clearly presented?

(x)

( )

( )

( )

Are the conclusions supported by the results?

(x)

( )

( )

( )

Comments and Suggestions for Authors

 Effect of individual rate of inbreeding, recent and ancestral in-2 breeding on wool quality, muscle conformation and exterior in 3 German sheep breeds

Keywords: inbreeding depression; sheep breeds; production traits; wool quality; muscling confor-56 mation; exterior; purging

There are words in the title as keywords. You can't repeat them. You must choose just one plaice.

Amended:

Line 56-57: inbreeding coefficient; ancestral inbreeding; breeding objectives; animal model; genetic parameters; genetic diversity; breeding directions

The abstract is very long and repetitive. The authors could be more objective and reduce this to a maximum of 250 words.

Amended:

Abstract shortened to 246 words.

The paper does not have a clear and defined objective.

Why were the objectives included in the Discussion? And not at the end of the introduction?

Line 353-355: Thus, the main objectives of the present study were to quantify the degree of inbreeding depression and in addition, whether negative variants may be eliminated through selection resulting in positive effects associated with ancestral inbreeding coefficients.

Amended:

Line 147-153:

The main objectives of the present study were to estimate genetic parameters for wool quality, muscle conformation and exterior and to quantify the extent of inbreeding depression and in addition, whether negative variants may be eliminated through selection resulting in positive effects associated with ancestral inbreeding coefficients. The present work thus allows an overall assessment of the influence of the breeding management practiced in the different sheep breeds from Germany and the resulting consequences with regard to inbreeding depression and purging.

Comments on the Quality of English Language

All work should be written in the third person. In some places, the authors say we (our)

Yes, indeed we use „we“ at some places. We feel this would be also appropriate as we have seen the use of „we“ in many papers.

Reviewer 2 Report

Comments and Suggestions for Authors

This paper reports results of analyses to determine whether increasing levels of different types of inbreeding (individual, ancestral, new etc.) are associated with decreased trait values used to select 30 sheep breeds in Germany. As currently presented though, it is very difficult to interpret the accuracy and relevance of the results because the authors have not properly defined most factors and variables investigated in their study. Rather, they simply assume that the reader has detailed knowledge of the different types of breeding directions for the 30 sheep breeds, the scoring systems used to select those sheep for the different breeding directions and even the different types of inbreeding being investigated in the study.

In particular, I have serious concerns about what appears to be a subjective scoring system for each of the three traits, where sheep failing the scoring system appear to be culled from the study, resulting in scores that are very similar across substantially different breeding directions (even though those breeding directions are not appropriately defined) and therefore the study lacks the variation that should be regarded as critical for a study of this topic.  

Until all these factors and variables are all appropriately defined, it is simply not possible to evaluate the relevance of the results as they are presented in the current draft of the manuscript. Additionally, in several places throughout the manuscript, the English language usage is very awkward, so it would be useful if a revised paper was checked for English language fluency. It is therefore suggested that major changes are needed to this paper before it is suitable for publication.

Specific comments are provided in chronological order below.

·             Line 14 and throughout the paper – what is the trait ‘exterior’? This sounds like some sort of visual appraisal of the sheep, but specifically, what traits are being evaluated and how is the trait actually scored?

·             Line 20 – delete ‘the’ before ‘estimate’

·             Throughout the paper – what is the definition of ‘new’ and ‘ancestral’ inbreeding? Presumably this relates to the number of recent or old generations, but given the study had access to pedigree records for a limited number of generations (some as few as 3.5 generations), it is important that these variables be accurately defined

·             Line 27 and throughout the paper – what are the ‘purging effects’ referred to here? This sounds like some sort of reduction in the rates of inbreeding or possible the effects of inbreeding, but a very clear and precise definition is required to enable accurate interpretations of the paper

·             Line 40 – change to read ‘All sheep breeds with sufficient data …”

·             Line 42 – presumably the heritabilities mentioned here are for each of the three traits examined in the study?

·             Line 44 – delete ‘the’

·             Line 75 – change ‘breed’ to ‘breeds’

·             Line 88 – insert ‘for’ between ‘selected’ and ‘in’

·             Lines 95 & 96, sentence beginning ‘Only …’ - awkward use of English

·             Line 108 – this line suggests that conformation may be the same as ‘exterior’ (questioned in the first dot point above), but that is then different to the usage in line 151 – hence, it is critical to accurately define what each of the traits is and how they are measured or scored (most of the traits indicated in the literature being cited are actually measured – e.g. weights in kg – but it seems that the traits recorded in the German sheep breeds may actually be subjectively assessed perhaps at a single time on numerous different attributes)

·             Line 121 and throughout the paper – what is ‘extinction’? The expression seems to suggest that inbreeding may no longer be a problem, but this could be due to low levels of inbreeding or resulting from outbreeding between completely unrelated animals – it is essential these terms are appropriately defined

·             Line 144 – what are ‘autochthonous’ breeds?

·             Line 151 – how exactly are the traits ‘wool quality, muscling conformation and ‘exterior’ assessed? This section suggests subjective scores on a 1-9 scale (though only sheep with a minimum score of 4 are used in this study), but no mention is made as to whether this is a single score for each of wool quality, muscling and exterior or whether for example, wool fineness, colour, length, density, balance and growth (lines 158-9) and ‘colour, shine, structure and balance of the coat’ are all scored independently by the assessor and then somehow combined into a single score describing wool quality. The same concern about the scoring system also applies to muscling and exterior. As currently described, it appears that the assessor simply subjectively scores the animals on all of the various aspects contributing to the trait as part of a single score, leaving lots of opportunities for errors around the individual aspects – and more importantly, resulting in scores across sheep that appear to lack variation that should be present in a study of this type

·             Models of analyses – as indicated above, the variables being analysed need to be appropriately defined e.g. what is the definition of ‘ancestral’, ‘new’, ‘classical’ and the ‘interaction with F’ inbreeding included in each of these models?

·             Lines 221-222 – each of these breeding directions need to be accurately described in the materials and methods section of the paper to allow the readers to understand what traits are important for the different breeding directions – and also to allow the readers to interpret the results and possibly why the results do not differ greatly for breeding directions that are generally expected to differ substantially (e.g. selection for wool and meat production would generally mean significant differences for wool quality traits and muscle conformation but that appears not to have occurred in this study, either because of problems with lack of variation in the scoring system that was used or because the breeding directions don’t vary in reality??)

·             Line 234 – what is an ‘upper and lower whisker’?

·             Line 249 – insert ‘and’ after 0.82

·             Line 254 – ‘significantly negative’ would be best placed between ‘was’ and ‘across’ to improve readability of this sentence

·             Results relating to Inbreeding Depression for wool quality traits (lines 303 and following) – it is very hard to interpret these results because of the lack of earlier definitions of both the scoring systems for the different traits and the breeding directions e.g. if wool and meat sheep were scored accurately, then it is expected there may be big differences between wool quality and muscle conformation traits for sheep with different breeding directions, but that is not the case, suggesting a problem exists, perhaps with the way the sheep are scored for these different traits

·             Without actually defining the breeding objectives, it is very difficult to interpret results for the different breeding directions e.g. HEO, EXO, CON, MON, HEA etc. as presented in these results (lines 303+) – though there is at least a more expected consistency of results for meat sheep and muscling (lines 315 and following)

·             Discussion section – quite a number of sentences in this section are actually repetition of results (e.g. lines 349-50, 353-355, 356-359, 360-365, 369-372, 373-377, 379-383)

·             Lines 388-89 – the fact that ‘phenotypic variances of the traits studied here were similar’ suggests there may have been a major problem with the scoring system for the different traits, rather than these results being accurate.

·             Lines 396-7 – again, the ‘similar size of the heritability estimates for all three traits’ may reflect a problem with the scoring system employed in the study, rather than these results reflecting reality

·             Lines 403-407 – rather than simply cite another study indicating studies have ‘been done’, it is important to discuss the results from those studies

·             Line 408 – are the traits being examined in the study being reported in this paper under ‘strong selection in breeding’ as suggested here as being ‘particularly important’ – without defining the breeding directions taken in this study, it is difficult to assess the relevance of this statement

·             Lines 425-426 – this information should have been provided in the introduction or materials and methods sections, when the breeding direction are defined and described

·             Lines 433-435 – this information would best be included in the introduction or materials and methods sections

·             Line 442 – replace ‘already found’ with ‘previously’

·             Line 447 – which experiment is referred to in this line – that of Wiener or the study being reported in this paper?

·             Lines 480 – 481 – which breeds are classified as threatened? These should have been described in either the introduction or materials and methods sections.

Comments on the Quality of English Language

As indicated in the previous comments, it is recommended that the English language use in the revised paper be checked prior to resubmission as there are currently very many awkward uses of English in the current draft paper

Author Response

Reviewer 2

We thank the reviewer for the valuable and useful comments in order to improve our manuscript.

We amended our manuscript accordingly.

Quality of English Language

( ) I am not qualified to assess the quality of English in this paper
( ) English very difficult to understand/incomprehensible
( ) Extensive editing of English language required
(x) Moderate editing of English language required
( ) Minor editing of English language required
( ) English language fine. No issues detected

Yes

Can be improved

Must be improved

Not applicable

Does the introduction provide sufficient background and include all relevant references?

( )

( )

(x)

( )

Are all the cited references relevant to the research?

( )

(x)

( )

( )

Is the research design appropriate?

( )

( )

(x)

( )

Are the methods adequately described?

( )

( )

(x)

( )

Are the results clearly presented?

( )

( )

(x)

( )

Are the conclusions supported by the results?

( )

( )

(x)

( )

Comments and Suggestions for Authors

This paper reports results of analyses to determine whether increasing levels of different types of inbreeding (individual, ancestral, new etc.) are associated with decreased trait values used to select 30 sheep breeds in Germany. As currently presented though, it is very difficult to interpret the accuracy and relevance of the results because the authors have not properly defined most factors and variables investigated in their study. Rather, they simply assume that the reader has detailed knowledge of the different types of breeding directions for the 30 sheep breeds, the scoring systems used to select those sheep for the different breeding directions and even the different types of inbreeding being investigated in the study.

Comments:

We agree that there is much more information needed to understand the rationale of breeding directions, scores used for breeding decisions and different inbreeding coefficients.

Amended:

Risk status: here we refer to Table S4 and our previous paper with Table S12: Autochthonous and imported sheep breeds in Germany with their risk levels, sustainable developmental goal (SDG) risk status, and their local endangerment status.

We added a supplementary Table S4 to describe the different breeding directions of sheep breeds.

In particular, I have serious concerns about what appears to be a subjective scoring system for each of the three traits, where sheep failing the scoring system appear to be culled from the study, resulting in scores that are very similar across substantially different breeding directions (even though those breeding directions are not appropriately defined) and therefore the study lacks the variation that should be regarded as critical for a study of this topic. 

Amended: 

We added the range of scores. All sheep presented for herdbook entry are registered with their final scores in OviCap. The range of all three scores is from 1-9. Therefore, these data of animals not selected for breeding in the herdbook are not discarded and the variation of all sheep intended for breeding is maintained.

Scores are specific for each breed because the typical wool quality, muscle and exterior of each breed has to be taken into account. This is very common in all livestock species and not a specific feature for sheep. The scores use the same range of values but the scores are centered on breed means (like estimated breeding values).

There are trained and very well experienced commissions performing the scoring. Therefore subjectivity should not be a major problem. In case of large subjective deviations, heritabilities should go to zero. This is obviously not the case. Similar estimates of heritabilities for the different final scores demonstrate that a similar variation for the different scores is achieved and this means that there is no shift or bias in any of these scores. Large differences would indicate that for the different final scores the scales would have been differentially used or animals would be stronger pre-selected for certain traits or selection intensity between the three final scores is very different. We can exlude this bias in our data.

Until all these factors and variables are all appropriately defined, it is simply not possible to evaluate the relevance of the results as they are presented in the current draft of the manuscript. Additionally, in several places throughout the manuscript, the English language usage is very awkward, so it would be useful if a revised paper was checked for English language fluency. It is therefore suggested that major changes are needed to this paper before it is suitable for publication.

 Amended:

We supplement required information. Please see below.

We revised the manuscript for use of English.

Specific comments are provided in chronological order below.

  • Line 14 and throughout the paper – what is the trait ‘exterior’? This sounds like some sort of visual appraisal of the sheep, but specifically, what traits are being evaluated and how is the trait actually scored?

Amended:

Line 199-240:

The responsibility for the performance tests of German herdbook animals lies with the breeding association and is carried out as a field test, either compulsory or voluntary depending on the breed, by trained personnel (commission of three experienced persons, including the head of breeding board) of the respective association. Overall scores for each of the three traits wool quality, muscling conformation and exterior are recorded for all breeds, for both male and female animals intended for breeding, within the framework of the herdbook entry according to the regulations in the herdbook. The scores awarded by the commission on a scale from 1-9 are valid for the whole life and serve as the basis for classification of animals by their breeding value and decision for herdbook entry. A minimum score of 4 must be achieved for each trait for herdbook admission. A score of 9 for wool quality, muscling and exterior is the optimum for an animal of each breed and corresponds best to the breeding objectives. The final scores of all animals presented at the field test are recorded in OviCap. Therefore, the whole variation of these final scores from 1-9 is captured for each breed. The final scores for wool quality, muscling conformation and partly for exterior (such as breed-typical pigmentation, body size and head shape) are breed specific and thus, centered on the breed average and therefore, not comparable between breeds. The evaluation takes place on selected locations for a group of animals from different farms. For the assessment of wool quality, a score from1-9 is given for the assessed criteria including fineness, colour, length, density, balance and growth at the shoulder, chest and haunches and then summarized with equal weights in a final score. In case of typical wool defects such as stalky, matted or yellow-sweaty wool, twist, colour defects or stitch hairs, the final score is downgraded to 1-4 depending on the grade and extension of the wool abnormality. Instead of the wool quality, the shedding behaviour during moulting and the coat quality (colour, shine, structure, balance) of hair sheep are evaluated, just as the wool quality, a score from 1-9 is given and combined to final score (Table S3). The evaluation should be done after the first wintering of the animals. For muscling conformation, the flesh-bearing parts of the chest, shoulder, back and haunch are evaluated and then combined into a final score for muscling, with the back and haunch being weighted higher. The following characteristics are the most important single criteria for exterior: head shape, dentition, horn status, head pigmentation and breed-typical appearance and pigmentation of the wool, legs and head, withers height, body length, rump depth, width of the breast, conformation of the neck and shoulder,  backline and conformation of the back, length, width and inclination of the pelvis,  angle and position of carpal and hock joints, angle at the pasterns, spreading of the claws and movement. The scores of the single criteria are summarized with equal weights in a final score if no defects in the body conformation are observed. In case of severe defects in body conformation such as underbite, kyphosis, lordosis, broken pasterns, steep ankle posture, or signs of disorders of sexual development like too small or split testicles, the final score is downgraded to less than 4 in order to exclude animals from breeding.

  • Line 20 – delete ‘the’ before ‘estimate’

Amended:

Line 20: employed to estimate the linear regression slopes

  • Throughout the paper – what is the definition of ‘new’ and ‘ancestral’ inbreeding? Presumably this relates to the number of recent or old generations, but given the study had access to pedigree records for a limited number of generations (some as few as 3.5 generations), it is important that these variables be accurately defined

Amended:

Lines 176-192:

The ancestral inbreeding coefficient according to Ballou (Fa_Bal) captures the proportion of the genome of an individual previously exposed to ancestral inbreeding [21]. Therefore, inbreeding in each ancestor is taken into account and summed up over all ancestors. A gradual increase in Fa_Bal compared to F over generations implies an increase in inbred ancestors in the pedigrees, but not necessarily an increase of the individual inbreeding coefficient.

Ancestral inbreeding according to Kalinowski et al. [22] (Fa_Kal) means that homozygous alleles have already met in former generations, whereas new inbreeding (FNew) is caused by alleles, which are identical-by-descent (IBD) for the first time in this pedigree. Fa_Kal is zero if classical inbreeding is zero because common ancestors have to be present on both sides of the pedigrees to be able to calculate an ancestral inbreeding coefficient Fa_Kal.

The concept of Ballou of inbreeding regards all ancestors in the pedigree unlike its contribution to the individual inbreeding resulting in an increase of Fa_Bal with an inbred ancestor in each generation of a pedigree. A faster increase of Fa_Bal compared to F across generations indicates an accumulation of inbred ancestors in the pedigrees, but these inbred ancestors do not necessarily contribute to the individual inbreeding coefficient.

  • Line 27 and throughout the paper – what are the ‘purging effects’ referred to here? This sounds like some sort of reduction in the rates of inbreeding or possible the effects of inbreeding, but a very clear and precise definition is required to enable accurate interpretations of the paper

Amended:

Line 193-198:

Purging reduces the negative effects of inbreeding depression, because many deleterious alleles express their harmful effects only when homozygous. Reducing the frequency of deleterious alleles in ancestral generations by natural or artificial selection after exposing them to inbreeding leads to purging. Purging is quantified by estimating the linear regression slope of ancestral inbreeding coefficients on the phenotypic trait. Positive estimates for ancestral inbreeding coefficients indicate the presence of purging.

The ancestral inbreeding coefficient according to Ballou (Fa_Bal) captures the proportion of the genome of an individual previously exposed to ancestral inbreeding [21]. Therefore, inbreeding in each ancestor is taken into account and summed up over all ancestors. A gradual increase in Fa_Bal compared to F over generations implies an increase in inbred ancestors in the pedigrees, but not necessarily an increase of the individual inbreeding coefficient.

  • Line 40 – change to read ‘All sheep breeds with sufficient data …”

Amended:

All sheep breeds with sufficient data were included resulting in 30 different breeds representing all breeding directions in different ecosystems.

  • Line 42 – presumably the heritabilities mentioned here are for each of the three traits examined in the study?

Amended:

Line 40-41:

Heritabilities were across all breeds of moderate size, with estimates at 0.18 for wool quality and muscling conformation, and at 0.18 for exterior.

  • Line 44 – delete ‘the’

Amended:

Line 41: Models employed to estimate

  • Line 75 – change ‘breed’ to ‘breeds’

Amended:

Line 97:

results of our previous study of genetic diversity of 35 different sheep breeds in Germany

  • Line 88 – insert ‘for’ between ‘selected’ and ‘in’

Amended:

Line 109-110:

provide insights into the effect of inbreeding depression on traits sheep are selected for in Germany.

  • Lines 95 & 96, sentence beginning ‘Only …’ - awkward use of English

Amended:

Line 117-118:

Very few previous studies reported similar traits that we study here.

  • Line 108 – this line suggests that conformation may be the same as ‘exterior’ (questioned in the first dot point above), but that is then different to the usage in line 151 – hence, it is critical to accurately define what each of the traits is and how they are measured or scored (most of the traits indicated in the literature being cited are actually measured – e.g. weights in kg – but it seems that the traits recorded in the German sheep breeds may actually be subjectively assessed perhaps at a single time on numerous different attributes)

Amended:

Lines 129-130:

results were available on the traits wool quality, muscling and exterior, so that our results again reflect a picture of

  • Line 121 and throughout the paper – what is ‘extinction’? The expression seems to suggest that inbreeding may no longer be a problem, but this could be due to low levels of inbreeding or resulting from outbreeding between completely unrelated animals – it is essential these terms are appropriately defined

Amended:

Line 142-143:

Depending on the extent of inbreeding and inbreeding depression, the probability of purging depends to a large extent on the genetic basis of the population in question,

  • Line 144 – what are ‘autochthonous’ breeds?

Amended:

Lines 170-172:

Autochthonous breeds according to German legislation are breeds that were first established in Germany or are only bred in Germany or breeds with an independent breeding programme in Germany.

  • Line 151 – how exactly are the traits ‘wool quality, muscling conformation and ‘exterior’ assessed? This section suggests subjective scores on a 1-9 scale (though only sheep with a minimum score of 4 are used in this study), but no mention is made as to whether this is a single score for each of wool quality, muscling and exterior or whether for example, wool fineness, colour, length, density, balance and growth (lines 158-9) and ‘colour, shine, structure and balance of the coat’ are all scored independently by the assessor and then somehow combined into a single score describing wool quality. The same concern about the scoring system also applies to muscling and exterior. As currently described, it appears that the assessor simply subjectively scores the animals on all of the various aspects contributing to the trait as part of a single score, leaving lots of opportunities for errors around the individual aspects – and more importantly, resulting in scores across sheep that appear to lack variation that should be present in a study of this type

Amended:

Lines 199-240:

The responsibility for the performance tests of German herdbook animals lies with the breeding association and is carried out as a field test, either compulsory or voluntary depending on the breed, by trained personnel (commission of three experienced persons, including the head of breeding board) of the respective association.

Overall scores for each of the three traits wool quality, muscling conformation and exterior are recorded for all breeds, for both male and female animals intended for breeding, within the framework of the herdbook entry according to the regulations in the herdbook.

A minimum score of 4 must be achieved for each trait for herdbook admission. A score of 9 for wool quality, muscling and exterior is the optimum for an animal of each breed and corresponds best to the breeding objectives.

The final scores of all three breeding objective traits and all animals presented at the field test are recorded in OviCap. Therefore, the whole variation of these final scores from 1-9 is captured for each breed. The final scores for wool quality, muscling conformation and partly for exterior (such as breed-typical pigmentation, body size and head shape) are breed specific and thus, centered on the breed average and therefore, not comparable between breeds.

For the assessment of wool quality, a score from1-9 is given for the assessed criteria including fineness, colour, length, density, balance and growth at the shoulder, chest and haunches and then summarized with equal weights in a final score. In case of typical wool defects such as stalky, matted or yellow-sweaty wool, twist, colour defects or stitch hairs, the final score is downgraded to 1-4 depending on the grade and extension of the wool abnormality. Instead of the wool quality, the shedding behaviour during moulting and the coat quality (colour, shine, structure, balance) of hair sheep are evaluated, just as the wool quality, a score from 1-9 is given and combined to final score (Table S3).

For muscling conformation, the flesh-bearing parts of the chest, shoulder, back and haunch are evaluated and then combined into a final score for muscling, with the back and haunch being weighted higher.

The following characteristics are the most important single criteria for exterior: head shape, dentition, horn status, head pigmentation and breed-typical appearance and pigmentation of the wool, legs and head, withers height, body length, rump depth, width of the breast, conformation of the neck and shoulder, backline and conformation of the back, length, width and inclination of the pelvis, angle and position of carpal and hock joints, angle at the pasterns, spreading of the claws and movement. The scores of the single criteria are summarized with equal weights in a final score if no defects in the body conformation are observed. In case of severe defects in body conformation such as underbite, kyphosis, lordosis, broken pasterns, steep ankle posture, or signs of disorders of sexual development like too small or split testicles, the final score is downgraded to less than 4 in order to exclude animals from breeding.

  • Models of analyses – as indicated above, the variables being analysed need to be appropriately defined e.g. what is the definition of ‘ancestral’, ‘new’, ‘classical’ and the ‘interaction with F’ inbreeding included in each of these models?

Amended:

Line 176-192:

The ancestral inbreeding coefficient according to Ballou (Fa_Bal) captures the proportion of the genome of an individual previously exposed to ancestral inbreeding [21]. Therefore, inbreeding in each ancestor is taken into account and summed up over all ancestors. A gradual increase in Fa_Bal compared to F over generations implies an increase in inbred ancestors in the pedigrees, but not necessarily an increase of the individual inbreeding coefficient.

Ancestral inbreeding according to Kalinowski et al. [22] (Fa_Kal) means that homozygous alleles have already met in former generations, whereas new inbreeding (FNew) is caused by alleles, which are identical-by-descent (IBD) for the first time in this pedigree. Fa_Kal is zero if classical inbreeding is zero because common ancestors have to be present on both sides of the pedigrees to be able to calculate an ancestral inbreeding coefficient Fa_Kal.

The concept of Ballou of inbreeding regards all ancestors in the pedigree unlike its contribution to the individual inbreeding resulting in an increase of Fa_Bal with an inbred ancestor in each generation of a pedigree. A faster increase of Fa_Bal compared to F across generations indicates an accumulation of inbred ancestors in the pedigrees, but these inbred ancestors do not necessarily contribute to the individual inbreeding coefficient.

  • Lines 221-222 – each of these breeding directions need to be accurately described in the materials and methods section of the paper to allow the readers to understand what traits are important for the different breeding directions – and also to allow the readers to interpret the results and possibly why the results do not differ greatly for breeding directions that are generally expected to differ substantially (e.g. selection for wool and meat production would generally mean significant differences for wool quality traits and muscle conformation but that appears not to have occurred in this study, either because of problems with lack of variation in the scoring system that was used or because the breeding directions don’t vary in reality??)

Amended:

Line 241-260:

In order to further investigate breeds with similar breeding objectives, we assigned the 30 sheep breeds to seven different breeding directions [4] with merinos, meat, country, mountain/stone, heath, milk and exotic breeds (Supplementary Table S4). Merino sheep have the finest wool and are fast-growing with medium size. Meat sheep are medium to large-framed, fast-growing and well muscled at the chest, shoulder, rump and back, they have also a good wool quality. Country sheep are well adapted to the ecosystem in which they evolved. Fineness and type of wool as well as muscling conformation are adapted to the local conditions. These sheep are undemanding, capable of marching and hardy, with a stable foundation and good claw health. Mountain and stone sheep are adaptated to the harsh mountain conditions in South Germany and have similar characteristics like country sheep. Another specific group of country sheep are the heath sheep. The heath breeds are small- to medium-framed animals that are indispensable for the landscape management on the heath and are particularly characterized by their frugality and robustness. They serve as versatile meat and wool suppliers and improve the fertility of the heath soils through their manure. The meat yield is very low as they are frugal, but the meat has a wildlife flavour. The dairy sheep considered in this study are large-framed East Friesian dairy sheep with moderate muscling and coarse wool. The exotic breeds are phenotypically rather different breeds such as hair sheep and small-sized sheep.

  • Line 234 – what is an ‘upper and lower whisker’?

Amended:

Line 326: he upper and the lower quartile

  • Line 249 – insert ‘and’ after 0.82

Amended:

Line 341:

positive value of 1.31±0.82, and this result was close to the significance threshold.

  •  

 Line 254 – ‘significantly negative’ would be best placed between ‘was’ and ‘across’ to improve readability of this sentence

Amended:

Sentence deleted.

  • Results relating to Inbreeding Depression for wool quality traits (lines 303 and following) – it is very hard to interpret these results because of the lack of earlier definitions of both the scoring systems for the different traits and the breeding directions e.g. if wool and meat sheep were scored accurately, then it is expected there may be big differences between wool quality and muscle conformation traits for sheep with different breeding directions, but that is not the case, suggesting a problem exists, perhaps with the way the sheep are scored for these different traits

Amended:

See also above. Scores are breed-specific like in all other livestock breeds. It is not possible to score all sheep breeds using a unifome scoring system.

  • Without actually defining the breeding objectives, it is very difficult to interpret results for the different breeding directions e.g. HEO, EXO, CON, MON, HEA etc. as presented in these results (lines 303+) – though there is at least a more expected consistency of results for meat sheep and muscling (lines 315 and following)

Amended:

Lines 208-210:

A score of 9 for wool quality, muscling and exterior is the optimum for an animal of each breed and corresponds best to the breeding objectives.

  • Discussion section – quite a number of sentences in this section are actually repetition of results (e.g. lines 349-50, 353-355, 356-359, 360-365, 369-372, 373-377, 379-383)

Amended:

Removed and moved to the Results section in lines 395428.

  • Lines 388-89 – the fact that ‘phenotypic variances of the traits studied here were similar’ suggests there may have been a major problem with the scoring system for the different traits, rather than these results being accurate.

Comment:

Similar estimates of phenotypic variances for the different final scores demonstrate that a similar variation for the different scores is achieved and this means that there is no shift or bias in any of these scores. Large differences would indicate that for the different final scores the scales would have been differentially used or animals would be stronger pre-selected for certain traits or selection intensity between the three final scores is very different. We can exlude this bias in our data.

  • Lines 396-7 – again, the ‘similar size of the heritability estimates for all three traits’ may reflect a problem with the scoring system employed in the study, rather than these results reflecting reality

Amended:

See above. Scoring is centered for each breed and in this case, we must expect similar phenotypic variances and also genetic variances. Otherwise, there are factors that bias these results. In all livestock breeds scores are breed-specific and scoring is done according to the variation within the respective breed. 

  • Lines 403-407 – rather than simply cite another study indicating studies have ‘been done’, it is important to discuss the results from those studies

Amended:

Line 543-553:

For example, Safari has done studies on genetic correlations and heritabilities of wool parameters (such as fiber diameter, staple length, fleece weight), growth (such as post weaning weight, conformation) and reproductive traits in Merino sheep [35,36]. For merino, a number of variance and heritability estimates were also reported for various wool quantity and quality parameters [37]. Heritabilities were also estimated for the Rambouillet breed on broadly comparable parameters [38]. The heritabilities for conformation, weaning and post-weaning from these previous studies [35,36,38] ranged from 0.05 to 0.23 and agreed well with the estimates for muscling conformation in the present study even if the traits were different. Fiber diameter showed heritabilities of 0.57-0.59 [36], 0.09-0.40 [37] and 0.05-0.22 [38], so the estimates from the present study are still in the same range.

  • Line 408 – are the traits being examined in the study being reported in this paper under ‘strong selection in breeding’ as suggested here as being ‘particularly important’ – without defining the breeding directions taken in this study, it is difficult to assess the relevance of this statement

Amended:

Line 554: inbreeding depression is particularly important for traits under selection in

  • Lines 425-426 – this information should have been provided in the introduction or materials and methods sections, when the breeding direction are defined and described

Amended: removed.

  • Lines 433-435 – this information would best be included in the introduction or materials and methods sections

Amended:

Line 571-575:

Due to the decreasing importance of wool quality in a large number of German breeds, there is an increased focus on muscling conformation and exterior. In contrast to theser traits, there are only a few breeds (5/30) with significant negative regressions for the individual rate of inbreeding on wool quality.

Comment:

This information is necessary to understand this part of the discussion.

  • Line 442 – replace ‘already found’ with ‘previously’

Amended:

Line 588:

traits were previously found in Wiener's work,

  • Line 447 – which experiment is referred to in this line – that of Wiener or the study being reported in this paper?

Amended:

Line 590:

With increasing inbreeding, Wiener reported a significant reduction in the size of body measurements,

  • Lines 480 – 481 – which breeds are classified as threatened? These should have been described in either the introduction or materials and methods sections.

Amended:

Line 258-260: Endangered breeds are mainly found under country, mountain/stone and heath sheep breeds (Supplementary Table S4). 

Comments on the Quality of English Language

As indicated in the previous comments, it is recommended that the English language use in the revised paper be checked prior to resubmission as there are currently very many awkward uses of English in the current draft paper

Amended: English language use in the revised paper has been checked.

Round 2

Reviewer 2 Report

Comments and Suggestions for Authors

The revised version of this manuscript is now much improved to enable the reader to better interpret the results and their implications. I am happy to recommend publication of the paper, with only a few minor suggestions to improve readability of the paper, as indicated in the following lines:

Definition of 'exterior' in the title and throughout the paper: although the term 'exterior' is now defined in the Materials and Methods section, its usage is not a term commonly applied to animal breeding. Hence, I am wondering whether the term 'visual appearance' might replace 'exterior' to enable easier interpretation by readers?

Line 12 - this line should read '... used to select sheep was investigated in all sheep breeds ...' (it appears from the tracked-changes version that 'was investigated' has been deleted

Line 21 - insert 'and' after 'rate of inbreeding' so the sentence reads '... rate of inbreeding and new and ancestral inbreeing'

Line 35 - replace 'on' with 'of'

Line 40 - replace 'at 0.18' with 'of 0.18'

Line 41 - change to read '... for all three traits' by deleting 'wool quality ... exterior'

Lines 41-42 - the sentence beginning 'Models employed ...' is difficult to understand, possibly because the word 'regarded' should be changed to 'dis-regarded'? Please check this sentence to ensure correct interpretation

Line 49 - suggest a brief definition of 'purging' be included in brackets near the first use of this term (e.g. reduction of negative effects of inbreeding depression in heterozygous forms)

Line 146 - delete 'an'

Line 152 - delete '6'

Line 537 - replace 'neglectable' with 'negligible'

Line 576 - it appears from the tracked-changes version that the number of animals may have inadvertently been deleted '(/30)'

Comments on the Quality of English Language

Minor changes are suggested to the use of some English terms in the suggestions above

Author Response

Reviewer 1

We thank the editors and reviewers for their valuable input, comments and recommendations to improve our manuscript. We revised our manuscript according to the comments and recommendations by the reviewer given. The minor changes as suggested by reviewer 1 on the use of some English terms were made according to the given suggestions. In addition, we have checked and corrected grammar, style and typing errors.

Open Review

Quality of English Language

( ) I am not qualified to assess the quality of English in this paper
( ) English very difficult to understand/incomprehensible
( ) Extensive editing of English language required
( ) Moderate editing of English language required
(x) Minor editing of English language required
( ) English language fine. No issues detected

Yes

Can be improved

Must be improved

Not applicable

Does the introduction provide sufficient background and include all relevant references?

(x)

( )

( )

( )

Are all the cited references relevant to the research?

(x)

( )

( )

( )

Is the research design appropriate?

(x)

( )

( )

( )

Are the methods adequately described?

(x)

( )

( )

( )

Are the results clearly presented?

(x)

( )

( )

( )

Are the conclusions supported by the results?

(x)

( )

( )

( )

Comments and Suggestions for Authors

The revised version of this manuscript is now much improved to enable the reader to better interpret the results and their implications. I am happy to recommend publication of the paper, with only a few minor suggestions to improve readability of the paper, as indicated in the following lines:

Definition of 'exterior' in the title and throughout the paper: although the term 'exterior' is now defined in the Materials and Methods section, its usage is not a term commonly applied to animal breeding. Hence, I am wondering whether the term 'visual appearance' might replace 'exterior' to enable easier interpretation by readers?

Comment: we kept the term exterior, but at the first usage we put „visual appearance of specific body parts“ in brackets. The term exterior includes more than visual appearence. Visual appearence does point in the direction of a more subjective impression of whole animal.

Line 12 - this line should read '... used to select sheep was investigated in all sheep breeds ...' (it appears from the tracked-changes version that 'was investigated' has been deleted

Amended

Line 12: used to select sheep was investigated in all sheep breeds

Line 21 - insert 'and' after 'rate of inbreeding' so the sentence reads '... rate of inbreeding and new and ancestral inbreeing'

Amended

Line 21: individual rate of inbreeding and new and ancestral inbreeding.

Line 35 - replace 'on' with 'of'

Amended

Line 25: current status of inbreeding depression

Line 40 - replace 'at 0.18' with 'of 0.18'

Amended

Line 30: with estimates of 0.18 for wool quality and muscling conformation,

Line 41 - change to read '... for all three traits' by deleting 'wool quality ... exterior'

Amended

Line 31: estimates of 0.18 for wool quality and muscling conformation, and of 0.14 for exterior.

Lines 41-42 - the sentence beginning 'Models employed ...' is difficult to understand, possibly because the word 'regarded' should be changed to 'dis-regarded'? Please check this sentence to ensure correct interpretation

Amended

Line 33-35: The models employed to estimate linear regression slopes for individual and ancestral inbreeding rates also account for non-genetic effects and the additive genetic effect of the animal.

Line 49 - suggest a brief definition of 'purging' be included in brackets near the first use of this term (e.g. reduction of negative effects of inbreeding depression in heterozygous forms)

Amended

Line 112: (reduction of negative effects of inbreeding depression due to selection for heterozygotes)

Line 146 - delete 'an'

Amended

Line 112: work on inbreeding purge

Line 152 - delete '6'

Amended

Line 119: muscle conformation and exterior and

Line 537 - replace 'neglectable' with 'negligible'

Amended

Line 448: seems to be negligible due to the very

Line 576 - it appears from the tracked-changes version that the number of animals may have inadvertently been deleted '(/30)'

Amended

Line 487-489: There were only a few breeds (4/30) with significantly negative regressions for the individual rate of inbreeding on wool quality, whereas for exterior and muscling conformation 11/30 and 14/30 reached significantly negative regression coefficients.

Comments on the Quality of English Language

Minor changes are suggested to the use of some English terms in the suggestions above

Amended as suggested.